# Claspin-Dependent and -Independent Chk1 Activation by a Panel of Biological Stresses

**DOI:** 10.3390/biom13010125

**Published:** 2023-01-07

**Authors:** Hao-Wen Hsiao, Chi-Chun Yang, Hisao Masai

**Affiliations:** 1Genome Dynamics Project, Department of Basic Medical Sciences, Tokyo Metropolitan Institute of Medical Science, Setagaya-ku, Tokyo 156-8506, Japan; hsiao-hw@igakuken.or.jp (H.-W.H.); yang-cc@igakuken.or.jp (C.-C.Y.); 2Department of Computational Biology and Medical Sciences, Graduate School of Frontier Sciences, The University of Tokyo, Kashiwa-shi, Chiba 277-8561, Japan

**Keywords:** Claspin, cellular stress, S phase, replication stress response, cell cycle

## Abstract

Replication stress has been suggested to be an ultimate trigger of carcinogenesis. Oncogenic signal, such as overexpression of CyclinE, has been shown to induce replication stress. Here, we show that various biological stresses, including heat, oxidative stress, osmotic stress, LPS, hypoxia, and arsenate induce activation of Chk1, a key effector kinase for replication checkpoint. Some of these stresses indeed reduce the fork rate, inhibiting DNA replication. Analyses of Chk1 activation in the cell population with Western analyses showed that Chk1 activation by these stresses is largely dependent on Claspin. On the other hand, single cell analyses with Fucci cells indicated that while Chk1 activation during S phase is dependent on Claspin, that in G1 is mostly independent of Claspin. We propose that various biological stresses activate Chk1 either directly by stalling DNA replication fork or by some other mechanism that does not involve replication inhibition. The former pathway predominantly occurs in S phase and depends on Claspin, while the latter pathway, which may occur throughout the cell cycle, is largely independent of Claspin. Our findings provide evidence for novel links between replication stress checkpoint and other biological stresses and point to the presence of replication-independent mechanisms of Chk1 activation in mammalian cells.

## 1. Introduction

Genome instability is a major driving force for cancer development [1]. Oncogenic stress has been shown to induce replication stress, which is the trigger for induction of genome instability. How oncogenic stress (e.g., Cyclin E overproduction) causes replication stress is still not clear, but the reduced levels of cellular nucleotide pool induced by oncogenic stress were shown to cause genome instability [1,2].

Living organisms are exposed to various types of stress, and are equipped with a variety of systems to manage them [2]. For example, in the cellular response pathway to replication failure in mammalian cells, the stress signal is transmitted from sensor kinase (ATR) to effector kinase (Chk1) to temporarily arrest progression of replication and cell division, and Claspin is involved in the ATR-Chk1 signaling axis in the replication stress response as an essential mediator [3,4,5,6,7].

Claspin and its yeast homologue, Mrc1, are essential for activation of downstream effector kinases (Chk1 and Cds1/Rad53, respectively) as replication checkpoint mediators [8,9,10,11,12,13]. Chk1 binding domain (CKBD) in metazoan Claspin was reported to be required for regulated Chk1 interaction [7]. It was also reported that Claspin could promote Chk1 phosphorylation at Ser317 and Ser345 in the presence of ATR in vitro [10]. Recently, we and others reported that either Cdc7 or CK1γ1 can phosphorylate CKBD of Claspin for checkpoint activation, though to different extents depending on cell types [3,14].

Cellular responses to environmental signals are important for cell proliferation and survival. Although detailed studies have been conducted on cellular responses induced by various types of stress, how these cellular responses cross-talk and control cell proliferation and survival in an integrated manner is largely unknown. Recently, it has been reported that DNA damage and/or Chk1 phosphorylation are induced by biological stresses, including ultraviolet (UV), arsenate (Ar), NaCl, lipopolysaccharides (LPS), hypoxia, heat shock, H_2_O_2_, and high glucose (HG) [15,16,17,18,19,20,21,22,23,24], suggesting the presence of cross-talks between various biological stress responses and replication checkpoint.

Ar and arsenite are derivatives from arsenic. However, due to the stronger cytotoxicity of arsenite than that of Ar, arsenite has been more extensively studied than Ar. Arsenite has been shown to interfere with DNA repair machinery and induce apoptotic cell death through regulating ATR, Chk1, and Chk2 signaling pathways in human cells [18,25,26]. Furthermore, 400 μM Ar treatment for 1 h has also been demonstrated to activate integrated stress responses through important eIF2α kinases in mouse embryonic fibroblasts (MEFs) [27].

In human cells, hypoxia was also reported to induce DNA damage and replication checkpoint activation through inducing expression of ATRIP, an activator of ATR [17,28,29]. It was also observed that hypoxia-activated unfolded protein response (UPR), which was sensed by PERK, IRE1, and ATF4 and that hypoxia partially blocked ongoing replication forks through PERK and decreased the capacity of new origin firing, suggesting that replication stress was generated by hypoxia [30,31,32,33,34]. Notably, the Claspin-Chk1 axis negatively regulates DNA replication during UPR. All the defective replication phenotypes triggered by hypoxia contribute to replication catastrophe potentially through inducing the expression of APOBEC3B, a DNA cytosine deaminase, further disrupting genome stability [35].

Furthermore, LPS treatment (1 ng/mL, 1 h) has been demonstrated to downregulate the gene expression associated with mitosis, DNA replication, DNA repair, and G1/S transition (e.g., Mcm2-5 and RAD51) in human and murine macrophages, and hypercapnia (high CO_2_ concentration; 20% CO_2_) was able to reverse this process [36]. Moreover, LPS treatment in combination with IL-4 induced Chk1 phosphorylation and DNA damage responses in B cells, although this may be due to the induction of CSR, which involves double-stranded DNA breaks [35]. Therefore, bacterial LPS is a potential agent that affects DNA replication and induces replication checkpoint [36,37].

Moreover, heat-induced Chk1 activation, which depended on Rad9, Rad17, TopBP1, and Claspin, was reported in human HeLa cells and chicken B lymphoma DT40 cells [38]. It has also been shown that the ATR-Chk1 axis is preferentially activated in HCT116 cells and Jurkat cells, a human T cell leukemia cell line, in response to heat shock (42–45 °C) and Chk1 inhibition in conjunction with heat shock can enhance apoptotic cell death [16,39].

In response to osmotic shock (NaCl), budding yeast Mrc1, homologue of Claspin, was phosphorylated by Hog1 kinase, and early-firing origins were delayed [40]. This response, however, does not involve Mec1 (sensor kinase) or Rad53 (effector kinase). Consistent with the finding in yeast [40], Claspin is directly phosphorylated by p38 MAP kinase, the mammalian homologue of Hog1 kinase, and safeguards cells from DNA damage elicited by osmotic stress in U2OS cells [19,20].

On the other hand, oxidative stress/H_2_O_2_ produced reactive oxygen species (ROS) and posed replicative threats by inducing replisome disassembly, stalling replication forks, and generating DNA breaks, comparable to the effect of HU in human cells [41,42]. Elevated ROS levels in response to H_2_O_2_ dissociated peroxiredoxin 2 (PRDX2) and Timeless from the chromatin, whose binding is critical for replication fork progression [42]. Moreover, the involvement of APE2, Apurinic/apyrimidinic (AP) endonuclease, which contains Chk1-binding motifs, is required for oxidative stress-induced Chk1 activation in a manner dependent on ATR in *Xenopus* egg extracts [24].

Additionally, HG condition can cause replication stress by provoking nucleotide imbalance in human cells. It introduces chemical modifications on DNA as a result of the adduct of anomalous glucose metabolism, giving rise to genome instability [22,23,43,44]. Interestingly, HG (37.8 mM glucose) compromised Chk1 activation and DNA damage response 1 h after UV irradiation or etoposide treatments, suggesting HG condition confers radio- and chemoresistance in cells. However, whether HG conditions alone activate Chk1 is not clear [22].

These above results strongly suggest that a wide spectrum of cellular stresses compromises progressing DNA replication forks by different mechanisms and that replication checkpoint, which is mostly ATR kinase-dependent, is activated by some of the stresses to maintain genome integrity [15,16,17,18,19]. However, how cellular responses to various biological stresses are linked to activation of replication checkpoint is largely unexplored [4].

We previously reported novel functions of Claspin in replication initiation and mechanisms of phosphorylation-mediated regulation of replication checkpoint activation [3,4,45,46]. More recently, we reported that Claspin regulates growth restart from serum starvation by activating the PI3K-PDK1-mTOR pathway [45]. Here, we have examined a potential role of Claspin in replication checkpoint activation in response to various cellular stresses and show that Claspin plays a crucial role in cellular responses to heat shock, hypoxia, arsenate, NaCl, oxidative stress, LPS, and HG. We show that some stresses suppress DNA replication and others do not have much effect on it. The Chk1 activation occurs throughout the cell cycle, but that outside the S phase is less dependent on Claspin than that within the S phase. We concluded that various biological stresses activate Chk1 either by direct activation of replication checkpoint in a Claspin-dependent manner or through distinct pathways that are independent of Claspin. The results also point to the presence of unknown mechanisms of Chk1 activation in mammalian cells.

## 2. Results

### 2.1. Various Biological Stresses Activate Chk1 and Induce DNA Damage

We examined the effect of various stresses on DNA damage and Chk1 activation by analyzing single cells through immunostaining (Figure 1A). The biological stresses chosen were, in addition to HU and UV (replication stresses), high temperature (heat stress), NaCl (osmotic stress), Ar (arsenate salt), LPS (bacterial infection), H_2_O_2_ (oxidative stress), HG (high glucose), and hypoxia (hypoxic stress).

We noted that all the biological stresses used induced Chk1 phosphorylation at S345 (pChk1(S345)) to different extents after 3 h treatment (Figure 1A, see also Appendix A). Under the same condition, γ-H2AX foci appeared in most cells exposed to these stresses, albeit to different extents. We also noted that some stresses (Ar, heat, H_2_O_2_) greatly reduced EdU foci, suggesting their inhibitory effects on DNA replication (Figure 1A; see also Figure 2).

These results indicate that Chk1 activation and DNA damage appear to be induced in most cells by any of the stresses. To determine the cell-cycle specificity of Chk1 activation and DNA damage more accurately, we next tried to quantify the fractions of γ-H2AX- and pChk1(S345)-positive cells in EdU-incorporating cells to access the S phase specificity of DNA damage and replication checkpoint activation induced by each stress. However, owing to strong inhibition of DNA replication by some stresses, it turned out to be difficult to accurately determine the relationship between cell cycle and DNA damage/Chk1 activation. Therefore, U2OS Fucci (fluorescent ubiquitination-based cell-cycle indicator) cells were treated with indicated stresses (Figure 1B–E). Fucci cells expressed two cell-cycle marker proteins, mKO2-Cdt1 and mAG-Geminin, marking G1-phase cells in red, cells in G1/S boundaries in yellow, and S/G2-phase cells in green [47]. We discovered that most stresses induced pChk1(S345) and DNA damage (γ-H2AX) throughout the cell cycle; however, to different extents (Figure 1B–E). Next, we quantified γ-H2AX-positive cells in each cell-cycle stage. γ-H2AX-positive cells were defined as cells with more than five foci of γ-H2AX. HU, Ar, NaCl, LPS, hypoxia and H_2_O_2_-induced γ-H2AX foci more preferentially in cells in G1/S transition and in S phase; whereas heat and HG also activated γ-H2AX in G1 phase to significant extents (Figure 1B,C). HU, hypoxia, and H_2_O_2_ induced pChk1(S345) foci preferentially in cells in G1/S boundaries and in S phase. Heat and HG induced pChk1(S345) foci in G1 cells more efficiently than in S phase cells, whereas Ar, NaCl, and LPS activated Chk1 in all the cell-cycle phases to a similar extent (Figure 1D,E). Strikingly, heat triggered pChk1(S345) foci formation in approximately 95% of G1-phase cells. Heat also induced γ-H2AX foci in more than 90% of the G1 cells. Similarly, fractions of G1 cells that showed pChk1(S345) and γ-H2AX signals under HG conditions were also higher than those of G1/S boundary and S/G2 cells, although the fractions and intensities of the signals were lower than those of heat-induced ones (Figure 1D,E). We also calculated mean fluorescent intensity (MFI) of γ-H2AX and pChk1(S345) under stresses. The results revealed that HU, Ar, and H_2_O_2_ induced stronger γ-H2AX MFI in G1/S boundary and S/G2 cells than in G1 cells, while NaCl, LPS, hypoxia, and HG exhibited similar levels of MFI of γ-H2AX throughout the cell cycle (Figure 1D). Heat not only induced γ-H2AX foci in more than 90% of the G1 cells but also showed higher MFI of γ-H2AX in G1 cells than that in cells in G1/S boundaries and in S/G2 cells (Figure 1D). Similarly, HU, Ar, and H_2_O_2_ showed more vigorous signals of pChk1(S345) preferentially in the G1/S boundary, and S/G2 cells, while NaCl, LPS, and HG treatments showed similar levels of the signal intensity of pChk1(S345) throughout the cell cycle (Figure 1E). Hypoxia exhibited stronger MFI of pChk1(S345) preferentially in G1/S boundaries compared to G1 cells and S/G2 cells (Figure 1E). Consistent with the result of cell numbers, heat exhibited higher MFI of pChk1(S345) in G1 cells, compared to G1/S boundary cells and S/G2 cells (Figure 1E). Taken together, we show that different cellular stresses activated Chk1 phosphorylation and DNA damage signals to different extents, and the response was also differentially regulated during the cell cycle. A notable conclusion is that all the stresses can activate Chk1 all through the cell cycle (Figure 1D,E, left graphs). Generally, intensity of Chk1 activation in G1 cells (red bars) is lower than that in G1/S/G2 cells (sum of yellow and green bars; Figure 1D and E, right graphs).

We next conducted FACS analyses to more accurately measure the DNA damage and Chk1 activation under different cellular stresses. Consistent with the results of immunostaining, HU and heat induced γ-H2AX foci in nearly 40% of all the cells and arsenate salt (Ar) in 23.6% of the cells, while other stresses (NaCl, LPS, HG, and hypoxia) induced γ-H2AX foci in approximately 11–16% (Figure 1F,G and Table 1).
We also analyzed pChk1(S345) and showed that HU, Ar, and heat activated Chk1 in 84.6%, 34.6%, and 32.7% of all the cells, while NaCl, LPS, hypoxia, and HG induced pChk1(S345) in a 17.7~27.1% population of all the cells (Figure 1F,G). These results indicate that Ar and heat induce Chk1 activation and DNA damage signals, while other stresses induce DNA damage and pChk1 signals at a lower level, consistent with the results of single cell analyses. We also analyzed RPA32 phosphorylation at S4/S8 (pRPA32), a marker of DNA damage. For the control cells without any treatment, 6.5% were pRPA32-positive, which could be due to spontaneous DNA damage during the ongoing DNA replication (Figure 1F,G). HU and heat induced pRPA32-positive cells in, respectively, 42.7% and 39.6% population of all the cells, and Ar 18.5%, while other stresses induced pRPA32 in only 6.13–11.4% populations (Figure 1G and Table 1). These results indicate different stresses induce DNA damage and replication checkpoint to different extents.

### 2.2. Biological Stresses Differentially Affect DNA Replication Fork Progression

To more precisely assess the effect of various biological stresses on DNA replication, we examined DNA synthesis in stress-treated cells by BrdU incorporation assay (Figure 2A). We treated cells with various stresses for 3 h and examined the cell cycle and BrdU incorporation by FACS. Consistent with the results of EdU imaging assay (Figure 1A), HU, Ar, heat, and H_2_O_2_ treatment for 3 h greatly decreased BrdU incorporation, whereas other stresses did not significantly affect the BrdU incorporation (Figure 2A). Cell-cycle profiles did not significantly change in HU, LPS, NaCl, and heat treatment. On the other hand, H_2_O_2_ treatment for 24 h led to increased G2 cell population, and UV treatment for 24 h led to significant cell death.

We next conducted DNA fiber assays to examine DNA replication fork progression and determined replication fork speed under different stress conditions (Figure 3). DNA was first labeled by CldU for 20 min, followed by IdU in the presence of various stresses for another 20 min (Figure 3A). This would permit us to examine the acute effect of stresses on DNA replication elongation. Twenty minutes of HU treatment almost fully activates Chk1. The ratio of IdU to CldU indicates the effect of the stresses on replication fork progression. The results showed that HU, heat, Ar, and H_2_O_2_ significantly retarded replication fork progression, consistent with the reduced DNA synthesis shown by FACS and BrdU incorporation (Figure 2A and Figure 3A). On the other hand, other stresses did not significantly impede fork progression, consistent with the results of BrdU incorporation. These results indicate that, in addition to HU, Ar, heat, and H_2_O_2_ also acutely inhibit DNA replication chain elongation. 

### 2.3. Effects of Stresses on Replication/Checkpoint Factors

We then examined the expression of various factors by Western blotting at 4 and 24 h after different stress treatments. HU and UV strongly induced pChk1(S345) at 4 h, while other stresses including heat, H_2_O_2_, NaCl, and LPS also induced pChk1(S345), albeit at a lower level. At 24 h after the exposure to heat, pChk1(S345) was reduced to the nonstimulated level, suggesting that cells might already had recovered from the stress or adjusted to the stress (Figure 4A). UV for 24 h also led to loss of pChk1(S345) signal, but this was due to cell death induced by UV (see next section). In contrast, pChk1(S345) was still detected at 24 h after treatment with HU, H_2_O_2_, NaCl, and LPS.

ATR, the upstream PIKK (Phosphatidylinositol 3-kinase-related kinase), is required for Chk1 activation. Phosphorylation of ATR at T1989 is an indicator of ATR activation. ATR was activated not only by HU and UV, but also by H_2_O_2_, salt, and LPS, albeit at a much lower level. Heat only slightly activated ATR at 4 h but not 24 h, similar to pChk1. Claspin undergoes phosphorylation upon replication stress (HU and UV), but also by other stresses, as exemplified by the mobility-shift on PAGE (Figure 4B). It appears that Claspin undergoes differential phosphorylation upon various stresses, as suggested by differential mobility shift (see also Figure 5D). RPA is phosphorylated at 24 h by heat and H_2_O_2_, suggesting the induction of DNA damage by these stresses.

### 2.4. Activation of Chk1 Kinase by Various Stresses Depends on Claspin

The results above convincingly show that various biological stresses can activate Chk1 phosphorylation. Mobility-shifts of Claspin induced by these stresses suggest activation of Claspin during the processes. Using Claspin ^f/-^ cells that we previously established, Claspin can be knocked out by infection of *Ad-Cre* viruses. By using this cell line, we analyzed the requirement of Claspin for Chk1 activation by various stresses. Consistent with the results from HeLa cells, not only HU- or UV-treatment but also various stresses including heat, H_2_O_2_, and LPS, induced pChk1(S345) in MEF cells. In accordance with the requirement of Claspin for efficient phosphorylation of Mcm by Cdc7, Mcm2 phosphorylation was reduced by Claspin knockout (Figure 5A). Chk1 phosphorylation, induced by various stresses, was reduced in Claspin knockout conditions (after *Ad-Cre* infection; (Figure 5A)). Treatment with 50 mM NaCl induced only a low level of Chk1 phosphorylation (Figure 5A, lane 13). ATR has been indicated to play a role in phosphorylation of S53 of Mcm2 [48]. This could explain the increased level of p-MCM2(S53) in response to some of the stresses (most notably by UV, heat, H_2_O_2_, NaCl, and LPS). This increased phosphorylation of MCM2 at S53 is decreased upon Claspin loss, reflecting the role of Claspin in replication checkpoint activation by various stresses. A 100 µM concentration of thymol (a phenol that is a natural monoterpene derivative of cymene and a volatile oil component) induced strong cell death, which was almost completely rescued by Claspin KO (Figure 5A, lanes 23 and 24), indicating that thymol-induced cell death of MEF cells depended on Claspin.

In HeLa cells, human cervical cancer cell line, the effects of Claspin siRNA on Chk1 activation by various stresses were examined. The same set of biological stresses activated Chk1 in a manner dependent on Claspin, although the levels of Chk1 activation were less than those achieved by HU or UV treatment (Figure 5B). Notably, Claspin was mobility-shifted by all stresses examined in HeLa cells, as was observed in MEF cells (Figure 5A), but the extent and patterns of the shifts varied, suggesting the induction of different phosphorylation patterns of Claspin by different stresses (Figure 5B–D).

We then examined the involvement of ATR, the upstream PIKK. ATR phosphorylation was induced by most of these stresses to differential extents, most notably by HU, UV, and H_2_O_2_ (Figure 5C, lanes 3–6, 11,12; see also Figure 4). ATR siRNA reduced Chk1 phosphorylation in cells treated with heat but the its effect with H_2_O_2_, NaCl, and LPS was not very strong, suggesting ATR is required for Chk1 activation by some of the stresses but not for others (Figure 5C, lanes 9–16). The weak dependency on ATR could be due to insufficient depletion of ATR. Alternatively, there could be other PIKKs that may be activated and transmit signals to Claspin by some stresses. In summary, Western analyses of Chk1 activation in the cell population indicate Claspin is required for Chk1 activation by these varieties of biological stresses, while ATR also plays a role for Chk1 activation at least by some of the stresses.

### 2.5. Roles of Claspin in Regulation of MAP Kinase Cascade and the PI3K-PDK1-Akt-mTORC1-4EBP1 Pathway

Next, we sought to investigate the role of Claspin in regulating MAPK pathways, which are activated by various environmental stress stimuli such as ultraviolet light, radiation, oxidation, heat shock, and hyperosmolarity, and induce cell death (apoptosis) in stressed cells [49]. In MEF cells, p38 MAPK or p44/p42 MAPK (ERK1/2), activated by MEK1/2 or MKK, respectively, was not affected by stresses or by Claspin depletion, except that UV treatment activated p38 MAPK (Figure 5A, lanes 17–32).

In HeLa cells, MAP kinases including p38 MAPK (Tyr180/Tyr182 phosphorylation), SAPK (stress-activated protein kinase)/JNK (Tyr183/Tyr185 phosphorylation), and p44/p42 ERK1/2 (Tyr202/Tyr204 phosphorylation) were activated by loss of Claspin (Figure 5B, lanes 1–18), whereas the protein levels of these MAP kinases were slightly reduced by Claspin KD. The stresses did not alter the levels of these phosphorylated proteins with or without Claspin siRNA, except that UV and thymol slightly activated p38 MAPK (Figure 5C, lanes 1–16).

Eukaryotic translation initiation factor 4E-binding protein 1, 4EBP1, is known to be phosphorylated by mTORC1 in response to growth stimulation, and this phosphorylation is required for its release from eIF4E and subsequent activation of cap-dependent translation. PDK1 kinase is activated by PIP3, resulting in activation of Akt and the PKC isoenzymes p70 S6 kinase and RSK. We recently reported that Claspin is required for activation of the PI3K-PDK1-mTOR pathway in response to serum activation [46]. Therefore, we examined if stresses affect this pathway.

Our results showed that T37/46 phosphorylation of 4EBP1 was not affected by any stresses or by depletion of Claspin in MEF cells (Figure 5A). In contrast, in HeLa cells, it was downregulated by Claspin knockdown, but not affected by any stresses examined. S241 phosphorylation of PDK1 was also inhibited by Claspin KD in HeLa cells, and was reduced by some stresses including HU, thymol, and 5FU. On the other hand, the Mcm2 phosphorylation (S53) was not affected in HeLa cells under the same condition, as reported before. Similar effects were observed in other cancer cell lines, including U2OS and 293T cells. Weak cell death was induced by some stresses including UV, thymol, heat, H_2_O_2_, and salt in the absence of Claspin in HeLa cells, as indicated by the cleavage of Caspase-3 (Figure 5B, lanes 24, 26, 28, 30, and 32). The level of Mcl1, a member of Bcl2 protein family associated with anti-apoptotic activity, was reduced by Claspin KD (Figure 5B).

Taken together, the results indicated that, in HeLa cells growing in the absence of stresses, Claspin plays suppressive roles in activation of the MAP kinase pathways, while it is required for activation of the PI3K-PKD-mTOR pathway.

### 2.6. Claspin-Dependent and -Independent Activation of Chk1 by Varieties of Biological Stresses

pChk1(S345) was induced by varieties of stresses not only in S phase cells but also in G1 phase cells (Figure 1B–E). We wondered if Claspin is required for Chk1 phosphorylation all through the cell cycle. To examine this, we used U2OS-Fucci cells and knocked down the expression of Claspin by siRNA, which was validated by Western blotting (Figure 6A). We quantified the fractions of cells showing pChk1(S345) signals under indicated cellular stresses in different cell-cycle stages (Figure 6B,C). We discovered that Claspin knockdown attenuated pChk1(S345) in S/G2 cells by 55 to over 70%, but it decreased pChk1(S345) in G1 phase cells only by 4 to ~30% under all the stress conditions except for NaCl (Figure 6D). With salt stress, Chk1 activation was downregulated by 40%, in G1 phase cells. The results indicate that various biological stresses activate Chk1 all through the cell cycle, but Claspin is required for Chk1 activation more predominantly during S phase.

## 3. Discussion

Cells are equipped with various stress response pathways that protect cells and living species from various environmental stresses. Among them, replication stress is mostly observed during S phase by varieties of treatment that impede progression of replication forks. Previous studies have indicated that “oncogenic stress” triggers cancer cell formation through inducing replication stress. Replication fork stalling caused by varieties of oncogenic stress generates DNA damage, which eventually leads to accumulation of genetic lesions, causing tumors to be formed. Although experimental “oncogenic stress” includes overexpression of Cyclin E, E2F, or growth factor receptors that can cause untimely growth stimulation, the nature of intrinsic “oncogenic stress” is rather unclearly defined.

### 3.1. Biological Stresses, DNA Replication, Chk1 Activation, ATR-Claspin, and other Signaling Pathways

We here provide evidence that diverse stresses, including oxidative stress (H_2_O_2_), heat shock, osmotic stress (high salt), and LPS, along with arsenate, high glucose, and hypoxia, can activate Chk1. It appears that these stresses can be classified into two categories (Table 2); one that arrests the replication fork and the other that does not obviously affect replication progression. The former may directly activate replication checkpoint, while the latter may indirectly activate it. We show that in both cases, Claspin is generally required for Chk1 activation in Western analyses of populations of the cells. We also showed that ATR may be required for Chk1 phosphorylation by these signals, although we cannot rule out the possibility that other PIKKs play a role.

It should be noted that there are some discrepancies between our results and other previous published studies. For example, our finding that hypoxia did not drastically impede replication fork progression was somewhat contradictory to studies that showed hypoxia significantly retarded S-phase progression [17,28,29,30,31,32,33]. Previous reports suggest inhibition of DNA replication by hypoxia treatment in RKO cells (poorly differentiated colon carcinoma cell line), but our DNA fiber and FACS analyses in U2OS or HCT116 showed no significant effect on replication fork progression or DNA synthesis (Figure 2 and Figure 3). This could be due to differences in the hypoxia condition. The concentration of oxygen was 0.5% for 20 min for DNA fiber and 3 h for FACS analyses in our experiments, in contrast to 0.1% for 8 h in the previous report. Inhibition of DNA replication by hypoxia may require duration of a low oxygen state for more than 3 h.

In our assays, some stresses (HU, Ar, heat, and H_2_O_2_) can efficiently arrest replication forks, whereas other stresses (NaCl, LPS, hypoxia, and HG) do not (Figure 2 and Figure 3), and generally, those stresses that inhibit DNA replication also induce DNA damage signals (γ-H2AX and pRPA32). A previous study in HeLa cells showed that heat treatment for 2 h in HeLa cells did not exhibit significant RPA32 phosphorylation [38]. Our Western analyses show also that RPA32 phosphorylation is detected at 24 h but not at 4 h after heat treatment (Figure 4). Thus, effects of various stresses on DNA replication and DNA damage could be affected by their strength and duration, together with the cell type used for the studies.

ATR activates two pathways: one leads to activation of Chk1 and the other to p38 MAP kinase [50]. Claspin is required for the former pathway, but not the latter. Claspin knockdown increased phosphorylation of MAP kinases including p38 MAPK, SAP1/JNK1, and ERK1/2 in cancer cells, suggesting it may negatively regulate the MAP kinase pathways during unperturbed growth. We also showed that Claspin is potentially required for activation of the PI3K-PDK1-mTOR pathway. We recently demonstrated that Claspin is required for growth restart of serum-starved cells, and this is due to its essential role for activation of the PI3K-PDK1-mTOR pathway [45]. Thus, Claspin may play a role for the activation of this essential signaling pathway during normal growth of cancer cells.

### 3.2. Chk1 Activation during S phase Depends on Claspin, but That in G1 Is Less Dependent on Claspin

We show here that a wide spectrum of cellular stresses activates Chk1 in a manner dependent on Claspin (Figure 7). Currently, it is not clear how Claspin is involved in Chk1 activation during stress-induced responses at the molecular level. Some stresses (Ar, heat, and H_2_O_2_) may acutely impede replication fork progression, and this may directly activate ATR-Claspin-Chk1. Others may not directly inhibit DNA replication, but Chk1 may be indirectly activated. By imaging and FACS-based analyses, we show that Chk1 activation in S phase depends on Claspin and that in G1 phase is largely independent of Claspin. In yeast, Mrc1, the Claspin homologue, and Rad9 are two mediator proteins that are required for checkpoint activation (phosphorylation of Rad53), though both act redundantly in Rad53 phosphorylation [51,52]. In yeast, Mrc1 is required specifically for S phase replication checkpoint, while Rad9 regulates checkpoint throughout the cell cycle. The roles of potential mammalian Rad9 homologue, 53BP1 or Mdc1, in Chk1 activation need to be evaluated.

Although single cell analyses indicate less dependency on Claspin for Chk1 activation in G1 phase, the population analyses of Chk1 activation by Western analyses show that pChk1(S345) in the presence of stresses is largely dependent on Claspin (Figure 5A,B). This is probably due to the fact that the level of Chk1 activation in G1 phase is generally lower than that observed in S phase (see right panel of Figure 1E; compare the red bar and the sum of the yellow and green bars).

Heat stress strongly inhibits DNA replication and also induces γ-H2AX signals in our experimental system. This finding leads to prediction that pChk1 and γ-H2AX signals predominantly appear during S phase. Indeed, HU, Ar, or H_2_O_2_, which inhibit DNA replication, induce these signals predominantly in S phase cells. In contrast, heat treatment induces them in more than 90% G1 cells in largely Claspin-independent manner. Similarly, HG activates Chk1 and γ-H2AX in G1 phase more efficiently than in S phase cells. The activation of γ-H2AX-pChk1 in G1 phase may reflect alteration of chromatin organization or epigenome state induced by the stresses, rather than DNA damage. Alternatively, aberrant transcription induced by stresses may generate RNA-DNA hybrids that may lead to DNA damage.

A previous study showed that Mrc1, the Claspin homolog in yeast, is phosphorylated through different SAPKs upstream of Mrc1, each of which responds specifically to different stresses, including osmotic, heat, oxidative stress, and low glucose [38]. In mammalian cells as well, different stresses can activate Claspin via different SAPKs upstream of Claspin [53,54]. Indeed, a recent report showed that osmotic stress induced Claspin phosphorylation by activated SAPK p38 and facilitated the repair of lesions in human cells [20]. We found that Claspin undergoes hyperphosphorylation in response to various stresses, suggesting different stresses may induce differential phosphorylation of Claspin, as indicated by the distinct shifted bands (Figure 5D).

Our findings indicate that various biological stresses activate Chk1 in both Claspin-dependent and -independent manners. They may directly interfere with DNA replication machinery or integrity of template DNA, or affects transcription profiles and chromatin state, ultimately generating sources for genomic instability. Activation of the effector kinase Chk1 may serve for protection of the genome from stress-induced lesions by modulating replication and cell-cycle progression. Further studies on cross-talks between cellular responses to various biological stresses and replication checkpoint pathway would reveal novel molecular mechanisms on how cells maintain genome integrity in the face of various environmental stresses and on how failures of cellular responses to stresses may lead to carcinogenesis.

## 4. Materials and Methods

### 4.1. Cell Lines

HeLa, U2OS, HCT116, and 293T cells were obtained from ATCC. *Claspin flox*/- mouse embryonic fibroblasts (MEFs) were established from E12.5 embryos [46]. *Claspin flox*/- MEFs stably expressing the wild-type or DE/A mutant Claspin were established by infecting recombinant retroviruses expressing these cDNAs [35]. Cells were grown in Dulbecco’s modified Eagle’s medium (high glucose) supplemented with 15% fetal bovine serum (NICHIREI), 2 mM L-glutamine, 1% sodium pyruvate, 100 U/mL penicillin, and 100 μg/mL streptomycin in a humidified atmosphere of 5% CO_2_, 95% air at 37 °C.

### 4.2. Antibodies

Antibodies used in this study are as follows. Anti-human Claspin was generated against the human recombinant Claspin with aa896–1,014 produced in *E. coli.* Anti-Chk1 phospho-S345 (#2348), anti-Chk1 phospho-S317 (#2344), anti-p44/42 MAPK (Erk1/2) (#4695), anti- SAPK/JNK (#9252), p38 MAPK (#8690), anti-p38 MAPK T180/Y182 (#4511), anti-p44/42 MAPK (Erk1/2) T202/Y204 (#4370), anti-SAPK/JNK T183/Y185 (#4668), Caspase-9 (#9508), Cleaved Caspase-3(#9661), and Mcl-1 (#5453) were obtained from Cell Signaling. Anti-α Tubulin (sc23948), anti-MCM2 (sc-9839), and anti-Chk1 (sc-8408) were obtained from Santa Cruz. Anti-phospho-H2A.X S139 (06-536) was purchased from Merck. Anti-BrdU (Ab6326) was purchased from Abcam. Anti-ATR phospho-T1989 (GTX128145) was purchased from GeneTex. Anti-BrdU (555627) was purchased from BD Pharmingen. Anti-H2A.X phospho-S139 (613402), anti-Rat IgG Alexa Fluor 555 (405420), and FITC-anti-BrdU (364104) were purchased from Biolegend. RPA32 phospho-S4/S8 (A300-245A) and anti-MCM2 S53(A300-756A) were purchased from Bethyl. Anti-Mouse IgG Alexa Fluor 488 (A-11017) was purchased from Invitrogen. Goat Anti-Rabbit IgG HRP (111-035-003) and Goat Anti-Mouse IgG (115-035-003) were purchased from Jackson ImmunoResearch Laboratory.

### 4.3. Claspin Knockdown by siRNA

Transfection of siRNA was performed using Oligofectamine™ Transfection Reagent (Invitrogen, Waltham, MA, USA) following manufacturer’s guidelines. All siRNAs were used at 20 pmol/mL. Transfections were performed for 48 h and cells were subjected to the indicated experiments.

siRNA sequences for Claspin siRNA were as follows [55]: siClaspin-nc#7 sense GCCAAUGAUCCUUCCUUCU-TT; siClaspin-nc#7 anti-sense AGAAGGAAGGAUCAUUGGC-TT.

### 4.4. Stress Conditions

To examine the stress responses in cancer cells, cells were treated with 2 mM hydroxyurea (HU), 50 J/m^2^ of UV, 100 μM Thymol, 42 °C (heat shock), 50 mM NaCl, 50 μM H_2_O_2_, 2 μg/mL *E. coli* lipopolysaccharides, 400 μM Arsenate salt (Ar), 4 °C (cold shock), DMEM with 30 mM glucose (high glucose), DMEM with 5.55 mM glucose (low glucose) or hypoxia (0.5% oxygen concentration in a CO_2_ incubator MG-70M (TAITEC)), respectively, for 3 h, unless otherwise stated.

### 4.5. Immunoblotting

To obtain whole cell extract (WCE), cells were first seeded in 12-well plates and cultured overnight. Exponentially growing cells were then treated with indicated biological stresses for 3 h at 37 °C. Cells were washed with PBS twice and directly resuspended by 1x sample buffer (Cold Spring Harbor Protocols). WCE was then run on 5–20% gradient SDS–polyacrylamide gel electrophoresis (PAGE; ATTO) and then transferred to Hybond ECL membranes (GE Healthcare, USA) followed by incubation with indicated antibodies. Signals were detected with Chemi-Lumi One Series for HRP (Nacalai, Kyoto, Japan) and images were obtained with LAS4000 (Fujifilm, Tokyo, Japan).

### 4.6. Flow Cytometry and Cell-Cycle Analysis

Cells were treated with indicated stresses and incubated with Bromodeoxyuridine (BrdU) at the final concentration of 20 μM for the last 15 min before the harvest. Cells were then washed and harvested. Cells were fixed with 4% PFA and incubated at 4 °C overnight. Cells then were then washed with PBS supplemented with 5% BSA and permeabilized and denatured by Triton X-100 (final concentration: 0.25%) and HCl (final concentration: 2N). Cells were then washed and all residual acid was neutralized with 0.1 M sodium borate for 2 min incubation. After washing, cells were then stained with anti-BrdU antibody conjugated with FITC and other primary antibodies diluted in wash buffer. Cells were stained with secondary antibodies at RT for 1 h. After washes, cells were then incubated with propidium iodide (PI) at RT for 30 min and samples were resuspended with PBS on ice and analyzed by flow cytometer BD LSRFortessa™ X-20. Data were then processed using FlowJo_V10 software.

### 4.7. Immunostaining

Fucci cells were treated with indicated stress conditions for 3 h and washed three times with PBS. Cells were fixed with 4% PFA in PBS for 15 min and then washed three times with PBS. After washing, cells were permeabilized with 0.5% Triton^®^ X-100 in PBS at RT for 20 min. After permeabilization, cells were blocked in 3% BSA/PBS for 1 h and indicated antibody staining. After staining, images were observed and analyzed by Zeiss LSM780.

### 4.8. DNA Fiber Assay

Exponentially growing cells were pulse labeled with 25 µM 5-Iodo-2′-deoxyuridine (IdU) at 37 °C for 20 min. Cells were then quickly washed three times with PBS and labeled by 100 µM CldU (5-Chloro-2′-deoxyuridine) at 37 °C for 20 min with indicated biological stresses. Cells were then incubated with 2.5 mM thymidine at RT for 30 sec after three quick washes with PBS. Cells were then trypsinized and resuspended with PBS at a cell density of 1 × 10^6^ cells/mL; 2 µL of labeled cells were mixed with unlabeled cells at the ratio of 1:1 and dropped onto the slides (Pro-01; Matsunami). The cell mixture was then lysed with the buffer (200 mM Tris-HCl and 50 mM EDTA with 0.5% SDS) for 5 min. Slides were tilted on the lid of a multiwell plate and DNA fibers flowed down along the slides at a constant speed. Fibers were then fixed with the solution containing methanol and acetic acid at the mix ratio of 3:1 at 4 °C overnight. Fibers were then denatured by 2.5 N HCl and blocked with PBS supplemented with 3% BSA and 0.1% Tween20. Samples were then stained with anti-BrdU antibody (Clone: BU1/75 (ICR1); Abcam) and anti-BrdU antibody (Clone: 3D4; BD) at RT for 1 h in the dark. After incubation with primary antibodies, fibers were then incubated with high salt buffer (28 mM Tris-HCl pH 8.0, 500 mM NaCl, 0.5% Triton X-100) at RT for 10 min in the dark. Fibers were then subjected to secondary antibody reactions and Hoechst staining at RT for 1 h in the dark. Fibers were visualized with Keyence BZ-X700 and quantified by Image J v1.53t.

## Figures and Tables

**Figure 1 biomolecules-13-00125-f001:**
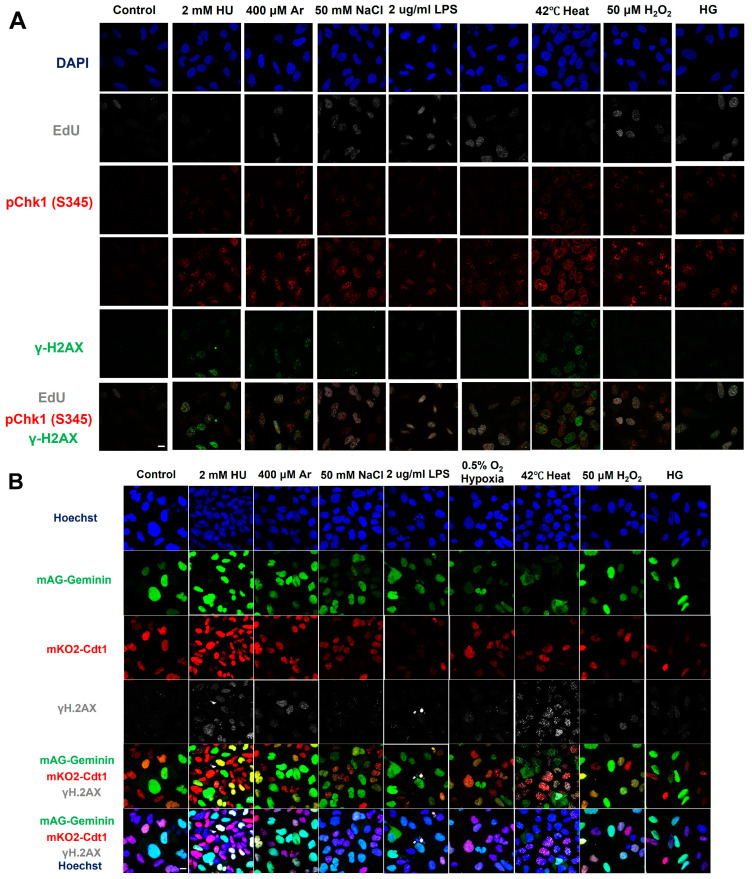
Differential effects on DNA replication and induction of DNA damage and replication checkpoint in a cell-cycle stage-dependent manner by various cellular stresses. (**A**) U2OS cells were exposed to indicated cellular stresses for 3 h, EdU-labeled for 15 min, and stained with indicated markers. Cells were then visualized and analyzed by confocal microscopy (Zeiss LSM780). Representative images are shown. Scale bar is 10 µm. Blue, DAPI (DNA); white, EdU (DNA synthesis); red, pChk1(S345) (replication checkpoint); green, γ-H2AX(DSB). A long-exposed version of pChk1 (S345) is also shown. (**B**,**C**) U2OS Fucci cells were exposed to indicated cellular stresses for 3 h and subjected to immunostaining. Cells were then analyzed by confocal microscopy (Zeiss LSM780). Representative images are shown. Scale bar is 10 μm. Blue, Hoechst (DNA); green, geminin (S/G2 marker): red, Cdt1 (G1 marker); white, γ-H2AX; yellow in the merged image, G1/S boundary. (**D**,**E**) Left: Fractions of U2OS Fucci cells containing γ-H2AX (**D**) and pChk1(S345) (**E**) foci were quantified for each cell-cycle population. Right: The mean fluorescent intensity of γ-H2AX (**D**) and pChk1(S345) (**E**) was quantified for each cell-cycle population. AU: arbitrary unit. (**F**) U2OS cells were exposed to different stresses for 3 h, and then were subjected to γ-H2AX (green), pChk1(S345) (red), and pRPA32 (S4/8) (phosphorylated single-stranded DNA binding protein representing DNA damage) staining, together with flow cytometry analyses. For γ-H2AX (green) and pChk1(S345) (red), cells were observed under confocal microscopy (Zeiss LSM780). Representative data and images are shown. Scale bar is 10 μm. (**G**) Quantification of the data from (**F**,**G**), fractions of γ-H2AX, pChk1(S345), or pRPA32 (S4/8)-positive populations are indicated for cells exposed to various stresses. Statistical analyses in (**D**,**E**) represent the mean values ± SEM under two independent experiments, all of which included three replicates (* *p* < 0.05, ** *p* < 0.01, *** *p* < 0.001, ns: no significant difference).

**Figure 2 biomolecules-13-00125-f002:**
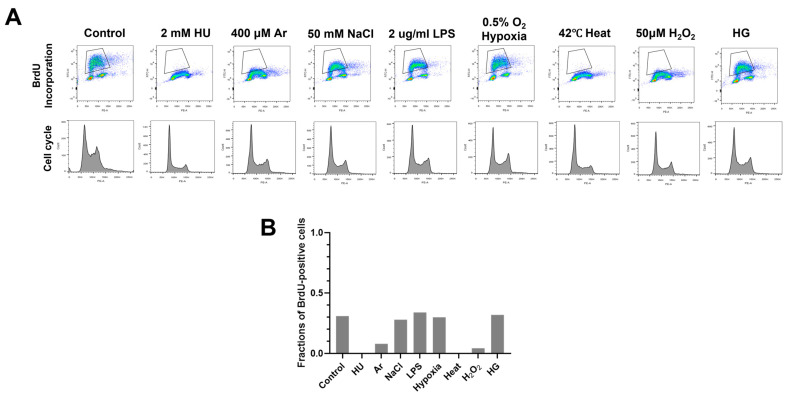
Various biological stresses influence DNA replication to different extents. (**A**) U2OS cells were treated with indicated biological stresses for 3 h. The nucleotide analog BrdU was added for 15 min before the cell harvest. Cells were then stained with anti-BrdU antibody and propidium iodide (PI); 1 × 10^4^ cells were analyzed by flow cytometry. Representative images were shown. Upper, BrdU incorporation (DNA synthesis); lower, cell cycle (DNA content). (**B**) Fractions of BrdU-positive cells in stress-treated cells (gated in A) were measured and are presented.

**Figure 3 biomolecules-13-00125-f003:**
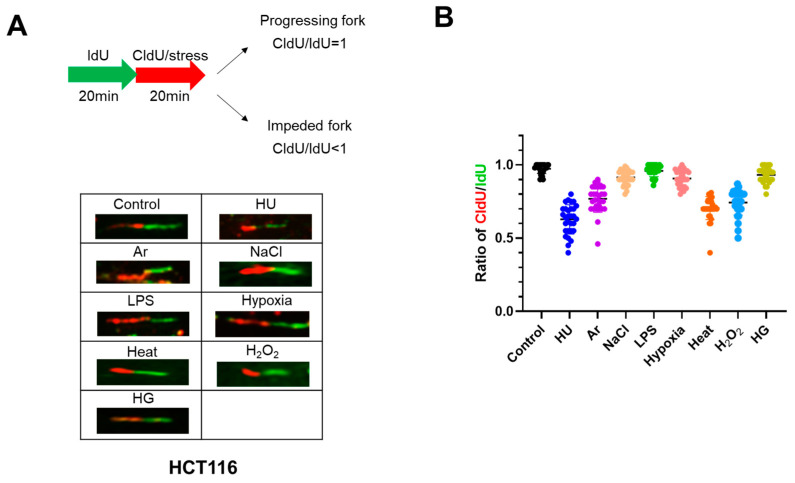
Different stress conditions differentially affect replication fork progression. (**A**) Scheme for the experiments for monitoring replication fork progression. Briefly, HCT116 cells were labeled with IdU for 20 min, followed by the labeling with CldU in the presence of stresses for another 20 min. The stress exposure time was short in order to detect acute effect of the stress on DNA replication. It has been known that exposure to HU for 20 min activates Chk1. The ratios of CldU/IdU less than 1 indicate that replication fork progression is impeded by the stresses. Representative DNA fibers under different stress treatments are shown below the scheme. (**B**) The CldU/IdU ratios in the presence of indicated stresses were determined and are presented. All statistical analyses represent the indicated mean values ± SEM under two independent experiments.

**Figure 4 biomolecules-13-00125-f004:**
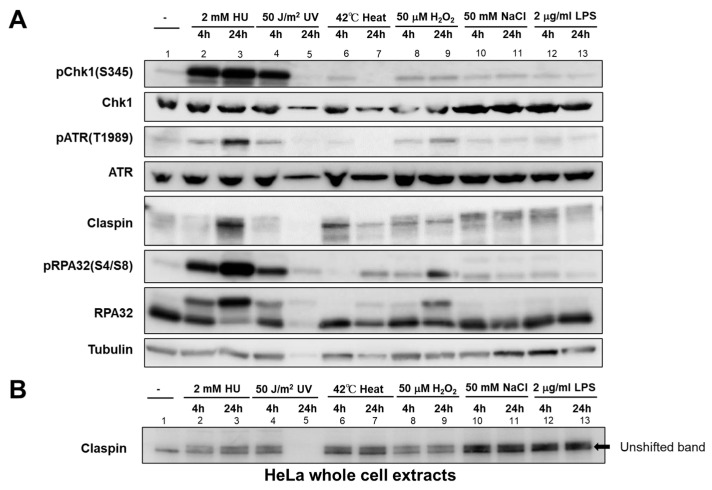
Effect of various biological stresses on checkpoint- and DNA damage-related factors in HeLa cells. HeLa cells were treated with indicated stresses for the time indicated. (**A**) The whole cell extracts were analyzed by Western blotting with the antibodies indicated. (**B**) Samples were analyzed on a low concentration gel to improve the separation of phosphorylated forms. The unshifted form of Claspin is indicated by an arrow. Lane 5, Protein is not detected due to cell death induced by UV.

**Figure 5 biomolecules-13-00125-f005:**
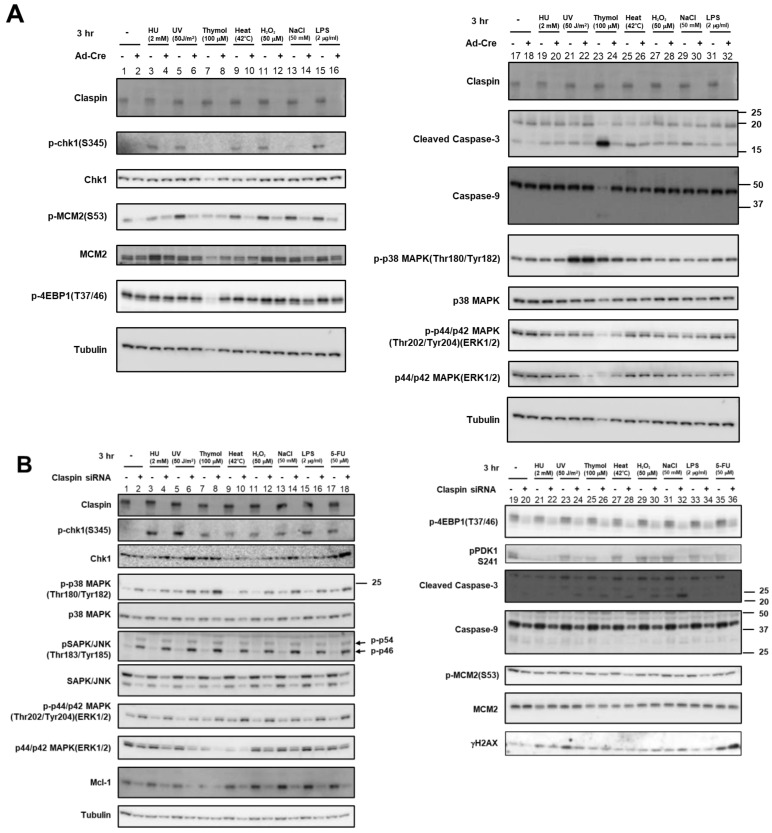
Effects of Claspin or ATR depletion on Chk1 activation by various biological stresses and on the factors involved in growth-related pathways. (**A**) Claspin(f/-) MEF cells were treated with Ad-Cre or nontreated and exposed to various stresses for 3 h. (**B**,**D**) HeLa cells were transfected with siRNA for Claspin (**B**) or for ATR (**C**), and were exposed to indicated stresses for 3 h before the harvest. The whole cell extracts were analyzed by Western blotting with antibodies indicated. SAPK/JUNK and p38 MAPK (MAPK activated by stresses and growth factors); ERK1/2 (MAPK activated by growth factors and mitogen); 4EBP1 (phosphorylated by mTOR and required for activation of translation); PDK1 (required for activation of mTOR); Mcm2 (phosphorylated by Cdc7 and required for replication); Mcl-1 (anti-apoptotic factor). Phosphorylated forms represent activated states. -, control siRNA. (**D**) Extracts prepared from HeLa cells treated with the stresses indicated were incubated in the absence of presence of λppase, and analyzed on a low concentration gel.

**Figure 6 biomolecules-13-00125-f006:**
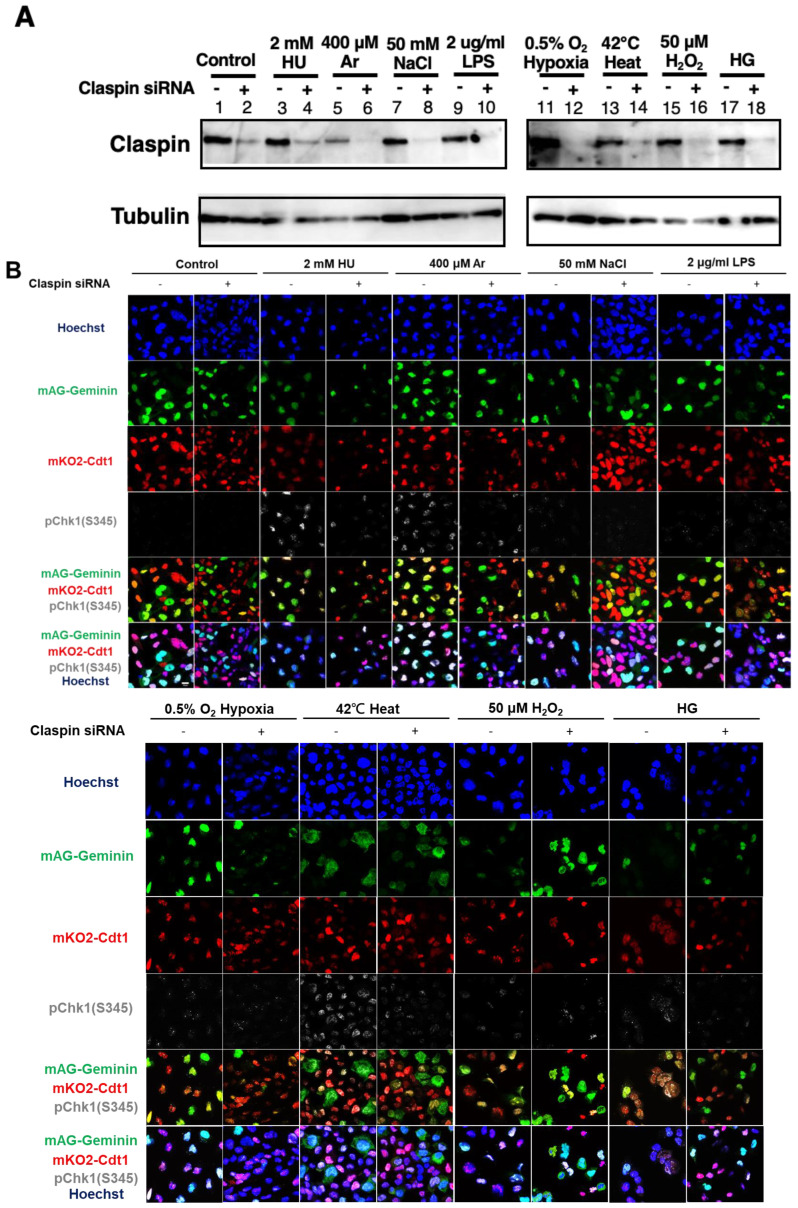
Claspin depletion abrogates Chk1 activation induced by various stresses mainly during the S phase. (**A**,**B**) U2OS Fucci cells were transfected with Claspin siRNA or with control siRNA for 48 h and were exposed to various stresses for 3 h before the cell harvest. The whole cell extracts from a portion of the cells were analyzed by Western blotting to detect Claspin and tubulin. (**B**) The same cells were observed under confocal microscopy (Zeiss LSM710). Blue, Hoechst (DNA); green, mAG-Geminin (S/G2 cells); red, mKO2-Cdt1 (G1 cells); white, pChk1(S345) (replication checkpoint). (**C**) Fractions of pChk1(S345)-positive cells in the U2OS Fucci cells of a specific cell-cycle stage after exposure to various biological stresses. (**D**) Ratios of pChk1(S345)-positive cells in Claspin-depleted versus control cells in cells of the specific cell-cycle stage. The higher values indicate the less dependency of the pChk1 signal on the Claspin function. All statistical analyses represent the indicated mean values ± SEM under two independent experiments, all of which included three replicates; + and—refer to the cells transfected with Claspin siRNA and those transfected with control siRNA, respectively.

**Figure 7 biomolecules-13-00125-f007:**
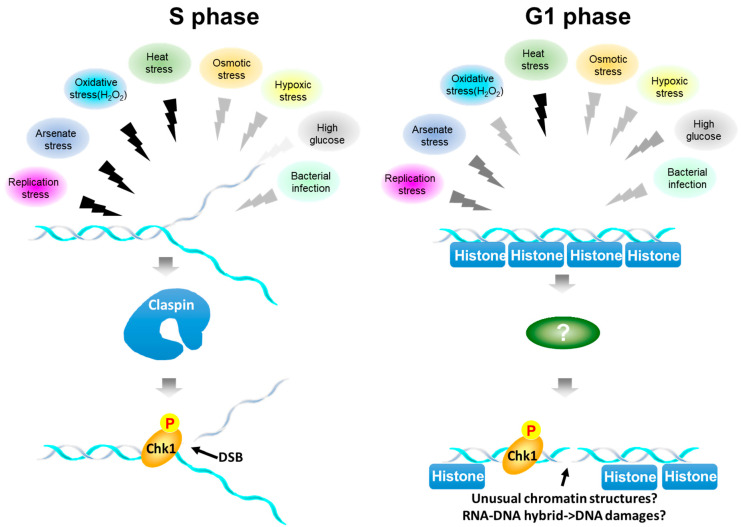
Summary of stress-mediated Chk1 activation during the cell cycle. Various biological stresses activate Chk1. Overall, Chk1 activation depends on Claspin. However, Chk1 activation is more strictly dependent on Claspin during S phase, while that in G1 phase is less dependent on Claspin. The colors of the zigzag lines represent the signal strength, black being highest and lighter gray being lower. During S phase (left), stresses with black arrows acutely inhibit DNA replication and may activate Chk1 and induce DNA damage at the stalled fork. Other signals also activate Chk1 albeit at a lower level. During G1 (right), all the signals can activate Chk1 to different extents. Heat, which strongly inhibits DNA replication, can vigorously activate Chk1 and DSB signal in G1 phase as well. During G1, γ−H2AX signals may represent actual DNA breaks or could be the results of reorganization of chromatin structures induced by stresses or those of RNA-DNA hybrid formation generated by stress-induced transcription, which may lead to DSB.

**Table 1 biomolecules-13-00125-t001:** Fractions of cells positive for BrdU, γ-H2AX, pChk1(S345), and pRPA32 in all the cell population in response to various biological stresses. The values are based on FACS data in Figure 2 and Figure 1F.

	BrdU-Positive Cells	γH2AX-Positive Cells	pChk1(S345)-Positive Cells	pRPA32(S4/8)-Positive Cells
Control	31%	4.08%	1.5%	6.5%
HU	0.1%	43.1%	84.6%	42.7%
Ar	7.8%	23.6%	34.6%	18.5%
NaCl	28%	11.3%	27.1%	7.77%
LPS	33.9%	12.0%	17.7%	9.12%
Hypoxia	30%	13.7%	21.1%	6.13%
Heat	0.6%	39.9%	32.7%	39.6%
H_2_O_2_	4.25%	60.7%	53.5%	31.2%
HG	31.9%	16.2%	20.4%	11.4%

**Table 2 biomolecules-13-00125-t002:** Summary of γ-H2AX, pChk1(S345), pRPA32, and BrdU in cells treated with various biological stresses. In the rows of “imaging”, the description is made on the basis of the data from the numbers of foci-positive cells (left graph of Figure 1D,E). “S > G1” indicates that foci are observed in S phase cells (green) more frequently than in G1 phase (red). γ-H2AX: +++ >40 % cells positive, ++ >20%, + >10%; pChk1(S345): +++ >50%, ++ >30%, ++ >10%; pRPA32: +++ >25%, + >4%; -, same withor below control.

Method	Signals	Control	HU (Replication Stress)	Ar (Arsenate)	NaCl (Osmotic Stress)	LPS (Bacterial Infection)	Hypoxia (Hypoxic Stress)	Heat (High Temperature Stress)	H_2_O_2_ (Oxidative Stress)	HG (High Glucose Stress)
Imaging	γH2AX (DNA damage)	S = G1	S > G1	S > G1	S > G1	S = G1	S > G1	S < G1	S > G1	S < G1
	pChk1 (S345) (replication stress)	S = G1	S > G1	S = G1	S = G1	S = G1	S > G1	S < G1	S > G1	S < G1
FACS	γH2AX (DNA damage)	-	+++	++	+	+	+	+++	+++	+
	pChk1 (S345) (replication stress)	-	+++	++	+	+	+	++	+++	+
	pRPA32 (DNA damage)	-	+++	++	+	+	-	+++	+++	++
	BrdU (replication)	+++	-	+	+++	+++	+++	-	+	+++

## Data Availability

The data presented in this study will be shared on reasonable request to the corresponding author.

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
