# Peer review of "Claspin-Dependent and -Independent Chk1 Activation by a Panel of Biological Stresses"

_biomolecules, 2023, doi:10.3390/biom13010125_

Round 1

Reviewer 1 Report

This paper is a comprehensive analysis of the effects of multiple biological stresses on various human/mouse cell lines. The focus is on DNA replication phenotypes. A large amount of data is summarised well and the authors make a convincing case that Chk1 activation is dependent on Claspin. The argument for the Chk1 pathway being S-phase specific is slightly less convincing given that the cell cycle categories were not phase specific (e.g. G1, G1/S, S/G2 rather than G1, S, G2) and the figure 6 control is variable. However the DNA combing data and author caveats ("predominantly", "largely") help with this.

All figures are clear and include appropriate controls. Additional marker explanations and justification of cell line choice (see specific comments) would help this paper to be understood by a wider audience.

Specific comments

55: Inconsistent text formatting

68: Malfunctioning?

Introduction: Ensure that the model organism is given each time. 

Throughout: There should be one space between digits and units e.g. 8 mM. This is inconsistent throughout the paper and absent in the figures.

Throughout: Unnecessary "the" and inconsistent plurals. This is particularly evident in the final paragraph of the discussion.

176-192 and Figure 1 F/G: It is odd to talk about these three measurements in a different order (γH2AX-pRPA32-pChk1) from the figure (pChk1-γH2AX-pRPA32). It might make sense to tackle γH2AX first followed by pChk1 since these have been the focus of previous panels. Alternatively the text order could be maintained and signposted to focus on DNA damage readouts first. Either way panel G could be integrated into panel F.

Introduction: Suggest introducing key phosphorylation sites of Chk1 so they are not new to the reader in the results section. Cite the literature which established them as key phosphorylation locations.

Table 1: Ctrl->Control to maintain consistency with figure 1, the word "control" is the same length as Hypoxia so it does not enlarge the table.

231: Give the range of time periods.

Figure 2: Please give an indication of n and reproducibility here. Perhaps the minimum n cell count would be appropriate?

246-7: Needs a title. This end of this sentence is unclear and should be rephrased.

250: Bdu->BrdU

Figure 3: Why has the cell line changed? 

256: less-><1 or less than 1.

261: Should there be a space after the /?

Figure 4: Why is UV now included as a stressor when it was not used in the earlier experiments?

309: This is a marginal observation for H2O2 and LPS. It is possibly visible without quantification for H202 (S317) but the bands look very similar in intensity for LPS.

310-1: Please justify this conclusion further.

Figure 5 C left panel: ATR should be on one line.

Figure 5: It would be helpful to explain what these proteins are markers/readouts for. This was done very well in the figure 1 legend.

Figure 5: Needs a title.

Figure 6: Please discuss the large variability of your control. Can you make any stressor comparisons given this variability?

Supplementary table 1: I found this summary table very useful and would encourage you to promote it to the main paper.

415: May be-> are some. There are definite differences.

417: The one-> studies. You cite multiple sources so there is not just one study with a contradictory result.

430: When was RPA32 phosphorylation assessed in this study? Could that explain the difference?

435: Is this point also supported by reference 48?

Figure 7 title: Suggest "Summary of stress-mediated Chk1 activation during the cell cycle".

Figure 7: 

465-6: "...could be the result of..."?

Methods: Please check phrasing of "washed with PBS for three times" -> washed three times with PBS?

588: Does Image J do calculation as well as quantification?

Reviewer 2 Report

The authors comprehensively demonstrated the effects of multiple stresses on the cellular ATR-CHK1 pathway and found that Claspin is essential for CHK1 phosphorylation at S/G2. However, this finding is not clearly stated in the title, abstract, and introduction sections. The logical structure of the article also needs to be optimized. The following are some specific issues:

1. In the introduction, the mechanisms by which various stresses cause DDR, replication stress, or activation of ATR are not involved in the manuscript. Please simplify it.

2. In Fig 1A, pCHK1 staining is too weak and it's difficult to be recognized, only except for the HEAT treatment. So it's hard to make a conclusion that the stresses induce CHK1 phosphorylation.

3. The evaluation of replication stress in fig 1A by EdU labeling is not reasonable, since the cells were not synchronized and many other factors could affect the staining. For an accurate result, a replication program should be applied, please check the paper (Aoolhnar Maya-Mhnggoza, Nature, 2018) for details.

4. In Fig 1F, the percentage of rH2AX+ cells in FACS seems much bigger than the number on the graphic. Please double-check.

5. In figure 3, the authors said that treatments and CldU were given together for 20 min. But I doubt that most stresses can cause significant damage in such a short period to slow down replication. Anyway, the fiber length is a reliable statistical indicator, not a ratio. Even untreated, DNA replication rates vary greatly, and the use of ratios only artificially reduces this variation in favor of statistical differences. In addition, the graphic of the HU treated sample is just a non-specific signal, not fiber.

6. Why did the author use different cell lines in each section? Does that mean the same result cannot reappear among the cell lines?

7. In fig 4, the author claimed a shift of Claspin and it's phosphorylation. However, there isn't a clear band in the control group. So, how to define the shift? Which band is unmodified?

8. The p-MCM2 (S139) is a signal of the initiation of replication. How about S53? Why MCM2 has a high level phosphorylation under some stresses?

9. In fig 5B, the shift of Claspin is still obscure. A low conc. gel may help. And a phosphatase treated control is also needed.

10. Why does the results section involve MAPK and metabolic pathways? How do they relate to replication stress, ATR-CHK1 signal, or the stresses used? The introduction section is not explicitly stated.

11. How can the inconsistent results of changes in claspin and p-p38 expression in M and H cells be explained? How does the inclusion of this section help the conclusions of the article, given that there are so many factors affecting the phosphorylation and expression of p38 as a stress sensor in cells and the lack of evidence in the study that claspin directly regulates p38 phosphorylation?

12. Please add the concentration of CldU in the methods.

Round 2

Reviewer 2 Report

My concern has been well addressed. I don't have any more questions.